# Role of Inositols and Inositol Phosphates in Energy Metabolism

**DOI:** 10.3390/molecules25215079

**Published:** 2020-11-01

**Authors:** Saimai Chatree, Nanthaphop Thongmaen, Kwanchanit Tantivejkul, Chantacha Sitticharoon, Ivana Vucenik

**Affiliations:** 1Faculty of Medicine and Public Health, HRH Princess Chulabhorn College of Medical Science, Chulabhorn Royal Academy, Bangkok 10210, Thailand; saimai.cha@cra.ac.th; 2Department of Physiology, Faculty of Medicine Siriraj Hospital, Mahidol University, Bangkok 10700, Thailand; nanthaphop94@gmail.com (N.T.); Chantacha.sit@mahidol.ac.th (C.S.); 3Sugavia Co., Ltd., Nakhonratchasima 30130, Thailand; ktantive@gmail.com; 4Department of Medical and Research Technology, School of Medicine, University of Maryland, Baltimore, MD 21201, USA; 5Department of Pathology, School of Medicine, University of Maryland, Baltimore, MD 21201, USA

**Keywords:** inositol phosphates, *myo*-inositol, IP6, energy metabolism, insulin resistance

## Abstract

Recently, inositols, especially *myo*-inositol and inositol hexakisphosphate, also known as phytic acid or IP6, with their biological activities received much attention for their role in multiple health beneficial effects. Although their roles in cancer treatment and prevention have been extensively reported, interestingly, they may also have distinctive properties in energy metabolism and metabolic disorders. We review inositols and inositol phosphate metabolism in mammalian cells to establish their biological activities and highlight their potential roles in energy metabolism. These molecules are known to decrease insulin resistance, increase insulin sensitivity, and have diverse properties with importance from cell signaling to metabolism. Evidence showed that inositol phosphates might enhance the browning of white adipocytes and directly improve insulin sensitivity through adipocytes. In addition, inositol pyrophosphates containing high-energy phosphate bonds are considered in increasing cellular energetics. Despite all recent advances, many aspects of the bioactivity of inositol phosphates are still not clear, especially their effects on insulin resistance and alteration of metabolism, so more research is needed.

## 1. Introduction

Inositol lipids and their derivatives, inositols and inositol phosphates (IPs), are well-known to be important to biology and signaling of eukaryotic cells [1]. *Myo*-inositol (myoIns) and inositol hexakisphosphate (IP6 or InsP6 or phytic acid) are common in biology; these naturally occurring carbohydrates are widely distributed among plants and mammalian cells, with multiple roles [2,3]. The broad spectrum of their actions has been shown to be related to the energy homeostasis, anti-oxidant and anti-inflammatory activities, and their role as neurotransmitters [4]. However, myoIns is only one of several possible structural isomers of inositol (1, 2, 3, 4, 5, 6-cyclohexanehexol) [2,5]. In the last two decades, myoIns dominated in scientific literature after it has been shown to successfully counteract cancer [4] and metabolic disorders including polycystic ovary syndrome (PCOS), gestational diabetes mellitus (GDM), infertility, and thyroid disorders [6]. However, recently the “other” inositols and inositol phosphates, present in both terrestrial and aquatic ecosystems, have received a lot of attention, and their biological role and medical applications have been indicated [7]. It is emerging that energy metabolism, and thus ATP production is closely regulated by these molecules. Therefore, in this review, we present the current knowledge on the numerous functions of these molecules and relate it to mammalian energy metabolism.

## 2. Biochemistry of Inositols and Inositol Phosphates

It is known that there are nine possible stereoisomers of inositol (a cyclohexanehexol structure) including *cis*-, *epi*-, *allo*-, *myo*-, *muco*-, *neo*-, (+)-*chiro*, (−)-*chiro*-, and *scyllo*-inositols [7,8,9]. These are formed through epimerization of its 6 hydroxyl group, and five of them—*myo*-, *scyllo*-, *muco*-, *neo*- and d-*chiro*-inositol—occur naturally, while the other four possible isomers (l-*chiro*-, *allo*-, *epi*-, and *cis*-inositol) are derived from myoIns [7,8,9]. Here, we illustrate all nine isomers of inositol in Figure 1.

Once, myoIns was considered to be part of the vitamin B family, later it has been known that it can be synthesized from sufficient amount of d-glucose, so it is not any more considered as a member of the vitamin B family [10]. Furthermore, myoIns is a cyclitol naturally present in animal and plant cells [4,11]. The pathway of myoIns synthesis from glucose 6-phosphate was revealed through 2 steps by the action of *myo*-inositol 1-phosphate synthase (MIPS) and *myo*-inositol 1-phosphatase converted glucose-6-phosphate to *myo*-inositol 1-phosphate and myoIns [12]. It can also be synthesized from the metabolism of inositol polyphosphates as discussed below. The phosphorylated forms of most of the inositol isomers are found in the environment, mainly in plants, although some of these isomers have been chemically synthesized [7,8]. Although it was originally thought that only 63 isomers were possible [13], today, 357 isomers have been identified from those 9 isomers [7], excluding the inositol pyrophosphates.

The cascade of inositol polyphosphates metabolism pathway in human cells, shown in Figure 2, is initiated by the formation of Ins (1,4,5) P3 from phosphatidylinositol bisphosphate (PIP2) by the enzyme phospholipase C (PLC) [14,15]. The intracellular Ins (1,4,5) P3 stimulates the endoplasmic reticulum (ER) resulting in Ca^2+^ release; however, most of the InsP3 is quickly dephosphorylated and consequently inactivated. While some of InsP3 is phosphorylated to 1, 3, 4, 5-tetrakisphosphate or Ins (1,3,4,5) P4 which will promote intracellular Ca^2+^ refilling and store from the extracellular fluid [4,16]. Conversion of Ins (1,4,5) P3 to Ins (1,3,4,5) P4 or Ins (1,4,5,6) P3 is mediated by inositol polyphosphate multikinase (IPMK). The other form of IP3, Ins (1,3,4) P3, is also known to exist in mammalian cells; however, its synthesis pathway is still unclear. Nonetheless, it can also contribute to the inositol polyphosphate pathway by getting converted to either Ins (1,3,4,6) P4 or Ins (1,3,4,5) P4 by the enzyme inositol-tetrakisphosphate 1-kinase. From there, all forms of InsP4 can be converted to Ins (1,3,4,5,6) P5 by inositol polyphosphate multikinase (IPMK), which interestingly is localized in the nucleus in high concentrations. Furthermore, inositol pentakisphosphate 2-kinase, which synthesizes IP6 from Ins (1,3,4,5,6) P5, is concentrated in the nucleolus, suggesting a need for IP6 within the nucleus [17].

In the inositol polyphosphates pathway, IP6 is a precursor molecule for inositol pyrophosphates (IPPs), complex biomolecules with at least one diphosphate group on one of the positions on the inositol ring. The four common forms of inositol pyrophosphates found in nature are 5-diphosphoinositol (1,3,4,6)-tetrakisphosphate(5PP-IP4), 1-diphosphoinositol (2,3,4,5,6) pentakisphosphate (1PP-IP5 or 1-IP7), 5-diphosphoinositol(1,2,3,4,6) pentakisphosphate (5PP-IP5 or 5-IP7) and 1,5-bisdiphosphoinositol (2,3,4,6) tetrakisphosphate (1,5PP2-IP4 or 1,5-IP8) [19]. The synthesis of 5PP-IP5 occurs by the conversion of IP6 by the enzyme inositol hexakisphosphate kinase 1 (IP6K1) [14,15]. Once 5PP-IP5 is formed, it can then be changed to 3,5PP-IP4 by diphosphoinositol-pentakisphosphate kinase 1. Conversely, Ins (1,3,4) P3 and Ins (1,4,5) P3 can be broken down into lower inositol phosphates and eventually myoIns.

IP6K1 is known to be a crucial enzyme affecting alterations of metabolism and metabolic diseases which were studied by genetic deletion experiments [20,21,22]. Inositol pyrophosphates generated from IP6 have been shown to increase cellular energetics by increasing glycolysis and mitochondrial function [23]. IP7 also plays a role in insulin signaling pathway by reducing insulin sensitivity in metabolic target organs including adipose tissues [21]. Among all inositol phosphates, IP6 seems to be the most intriguing agent for various reasons. A six-carbon inositol ring in IP6 represents the basic moiety of carbohydrate and its lower phosphate derivatives (IP1-5); IP6 is a very stable and the most abundant polyphosphate in nature [24,25]. It is a component of cereal diets and legumes, found in rice, wheat, peas, beans, oats, barley, in concentrations ranging from 0.4–6.4%, where it is referred to as phytic acid, making it readily available for consumption [24,25]. Based on the aforementioned pathway, the 2-position is the last position to be phosphorylated in mammalian cells and the presence of the phosphate group at positions 1,2,3 (axial, equatorial, and axial) contributes to its unique properties as an antioxidant and specific chelating capacity of potentially toxic elements [26,27]. The unique structure of IP6 is illustrated in Figure 3A. Introduction of Agranoff’s turtle analogy helps to visualize *myo*-inositol hexakisphosphate in the form of turtle [13], in which the head represents the axial hydroxyl group, and the five equatorial hydroxyls serve as forelimbs, hind limbs, and the tail [13,28], as illustrated in the Figure 3B. This particular orientation may be advantageous for electrostatic interaction with protein domains [29].

Although much of what we know about the inositol phosphates metabolism was studied in *Dictyostelium discoideum* and *Saccharomyces cerevisiae*, the biochemical property of inositol phosphates was first revealed by Mabel R. Hokin in 1953, showing that the incorporation of P^32^ into phospholipides enhanced acetylcholine activity to increase amylase secretion in pancreas slices of pigeons [30]. This finding explored the relationship between accelerated phospholipide synthesis and specific enzyme activity [30], suggesting a special function of inositol phosphates. Interest in the inositol phosphates gained momentum in the late 1980s. Later in 1991, Europe-Finner et al. revealed that different patterns of [^3^H]-inositol incorporation into inositol phosphates elicited different response than the phosphatidylinositol phosphates during the development of *Dictyostelium discoideum* [31]. The incorporation into IP6 was very rapid and increased linearly over 8 h, which was 20 times faster and accumulated 30–50-fold higher compared to that of Ins (1, 4, 5) P3. Yet, the inositol polyphosphates precursors only occurred 3 h after development. These data suggest the possibility of a metabolic switch, rather than stimulation to the threshold of its precursor, is required during development [31].

## 3. Biological Roles and Activities

Although signaling via inositol phosphates, e.g., the second messenger *myo*-inositol 1,4,5-trisphosphate, and phosphoinositides, is well documented in biological processes, myoIns and the other inositol phosphates also possess biological activities [7]. It is well-known that plant cells contain myoIns either in its free form (as inositol-containing phospholipids or phosphoinositides) or as phytic acid (IP6), a principal storage form of phosphorus in plants, particularly in bran and seeds [10,24]. High amounts of myoIns are found in various vegetables, fruits, beans, nuts, grains and milk—for example, in green shelled beans, artichoke, okra, cantaloupe, grapefruit, and lime [11]. Likewise, almost all mammalian tissues and cells contain high concentrations of IP6, myoIns, and other inositol phosphates, including liver, brain, kidney, and lung of rats as well as HeLa cells, human erythrocytes, and human white blood cells [32]. These higher inositol phosphates may have valuable role because of the nature of their high electrostatic force. Thus, they can act as cofactors or intermolecular glue [33] that bring proteins together to activate various biological processes, including RNA editing [34], RNA export [35,36], mRNA transcription [35], DNA double stranded break repairs [37,38], gene expression [39,40,41], proteasomes [42,43], and phosphate homeostasis [44], some of which are discussed below.

*RNA editing**.* As mentioned above, the enzymes IMPK and IP2K, which are necessary to synthesize IP6 from its precursor, are concentrated in the nucleus, suggesting a role for IP6 in nuclear function in human cells. Indeed, crystal structures have revealed interactions of IP6 with various regulatory elements through electrostatic interaction with proteins containing highly basic residues. Macbeth et al., reported in 2005 that IP6 binds to the core catalytic domain at the core of adenosine deaminase (ADAR2), an RNA editing enzyme that converts adenosine (A) to inosine (I) [34].

*RNA export and gene expression.* In eukaryotes, messenger RNA needs to be processed to messenger ribonucleoprotein (mRNP) in order to be exported from the nucleus to the cytoplasm through nuclear pore complex (NPC), which requires the action of the DEAD-box ATP-dependent RNA helicase DDX19. However, the recruitment and interaction of nucleoporins Nup42 and Gle1 is necessary for activity [45]. It has been shown in several studies that IP6 mediates and enhances the activities of DDX19 through its interaction with Gle1 [35,46]. Furthermore, siRNA knockdown of *IPPK* gene revealed selectivity for a subset of RNA involved, including those that affect cell cycle G1/S checkpoint regulation and inflammatory response [35]. Another study showed that depletion or catalytic inactivation of IPMK inhibits the export of homologous recombinant factors, such as RAD51 and CHK1 [37]. However, IP6 binding to non-homologous end-joining repair protein Ku has been demonstrated in pulldown assays [47]. In addition, IP6 depletion resulted in decreased Ku mobility [38], suggesting the role of these higher inositol phosphates in double stranded break repair mechanisms as well [47,48]. In contrast, Ins(1,4,5,6)P4 preferentially interacts with histone deacetylase proteins HDACs to regulate gene expression [39,40,41].

*Phosphate homeostasis.* Using CRISPR to completely disrupt *IP6K1* and *IP6K2* genes in a human colon carcinoma HCT116 cell line, it was shown that although there was an increase in the amount of ATP and intracellular free phosphate, these cells had limited ability to uptake radioactive phosphates [44]. This suggests that the pyrophosphates have a role in cellular phosphate homeostasis. Furthermore, the interaction between inositol pyrophosphate 1,5PP-IP4 with XPR1 can occur [49]. Since XPR1 is the only protein known mammalian protein involved in phosphate export; thus, its interaction was investigated. Specifically, XPR1 contains SPX domain that has been known to interact with inositol polyphosphates, which was confirmed in various knock out models that the interaction between XPR1 and PPIPs can regulate mammalian cell phosphate homeostasis [44,49,50].

## 4. Effects on Insulin Resistance and Energy Metabolism

### 4.1. Basic Pathophysiology of Insulin Resistance

The prevalence of diabetes mellitus has been growing worldwide with an estimated to affect 9.3% or 463 million people in 2019 which will be increasing to 10.2% or 578 million by 2030 and 10.9% or 700 million by 2045 [51]. Diabetes mellitus also leads the development of comorbidities including cardiovascular diseases, chronic kidney disease, and eye damage as well as declined quality of life [52]. Insulin resistance has been considered to be associated with obesity, type 2 diabetes, and cancer [53,54,55]. The pathophysiology of insulin resistance is the defects of insulin receptors in response to sustained hyperinsulinemia observed in obese and/or diabetic patients [54]. Blood insulin levels were shown to be high after glucose infusion in normal weight subjects while its levels seemed to be impaired in obese and diabetic patients [56]. Moreover, a study found that body mass index (BMI), waist circumference, and body fat percentage had positive correlations with insulin resistance in overweight and obese human subjects [57]. Blood glucose level and glycemic status might determine blood insulin levels, pancreatic islet β-cell function, and insulin resistance status [58]. Insulin resistance leads to β-cell compensation, exhaustion, and consequently dysfunction of β-cell resulting in impaired insulin secretion [58]. A study in mice found that β-cell dysfunction induced by forkhead box protein O1 (FoxO1) ablation resulted in hyperglycemia, loss of β-cell mass, β-cell demise, decreased plasma insulin levels, and decreased pancreatic insulin levels [59] suggestive of the progressive decline of β-cell function inherently associated with the development of both type 1 and 2 diabetes mellitus. This pathology will lead to increased plasma glucose levels produced from liver and muscle as well as increased free fatty acid from adipocyte lipolysis, resulting in increased insulin resistance [54], occurring like a vicious cycle.

### 4.2. Inositols, Inositol Phosphates and Insulin Resistance

In 1986, the role of inositols in insulin signaling pathway was demonstrated by a research group showing that insulin and phosphatidylinositol-specific phospholipase C enzyme (PLC) synergistically increased cyclic nucleotide (cAMP) phosphodiesterase (PDE) activity in fat cells of rats [60]. PDE is an enzyme known to breakdown phosphodiester bonds, control the phosphorylation process, and regulate protein–protein interactions [61]. So, it might be implied that PLC might enhance insulin actions through cAMP-dependent pathway in the fat cells. Later in 2003, a study found that both low and high doses of buckwheat containing high levels of d-*chiro*-inositol decreased serum glucose in diabetic rats [62]. This evidence possibly triggered the interest in the association between inositols and insulin action in recent years. For example, studies found that myoIns and d-*chiro*-inositol have been shown to counteract metabolic disorders [63] and improved insulin resistance [63,64,65].

MyoIns is the most common isoform shown to have therapeutic effects in human health including metabolism, reproduction, cell growth and survival, and development of nervous system [10]. Moreover, studies revealed that inositol phosphates and myoIns combined with d-*chiro*-inositol, or myoIns alone improved lipid profiles including decreased plasma low density lipoprotein (LDL) and triglycerides and increased plasma high density lipoprotein (HDL) [63], decreased hemoglobin A1C (HbA1c) [66], reduced blood glucose levels [62,63,66,67], and decreased the homeostatic model for assessment of insulin resistance (HOMA-IR) which is an index for insulin resistance [63] in diabetic/PCOS patients and/or rats. Basically, HOMA-IR is obtained by calculation of the following formula: HOMA-IR = (fasting plasma insulin (μU/mL) x plasma glucose (mmol/L))/22.5 [68]. Furthermore, myoIns decreased jejunum glucose absorption and increased muscle glucose uptake in normal rats, as well as decelerating gastric emptying and accelerating digesta transit in diabetic rats [67]. Moreover, a study showed that myoIns and d-chiro-inositol decreased insulin levels after oral glucose tolerance test (OGTT) in obese human subjects [64], suggesting that it can induce insulin sensitivity. MyoIns deregulation has been found in numerous conditions mechanistically and epidemiologically associated to a high-glucose diet or altered glucose metabolism [55,69]. Its insulin-mimetic properties have been found to be efficient in lowering post-prandial blood glucose and associated human disorders [10]. Targeting insulin resistance, myoIns has been effective in gestational diabetes mellitus [10], metabolic syndrome [10], and PCOS [10,70]. A meta-analysis performed in PCOS patients revealed that myoIns supplementation improved metabolic profiles including blood insulin and HOMA-IR index [71]. Interestingly, myoIns was able to modulate both insulin resistance and cancer, by targeting multiple biochemical processes that are shared in both cancer and insulin resistance-based diseases [55].

IP6 has been shown to reduce blood glucose and delay carbohydrate digestion and absorption after supplementation in humans [72]. A study in rats found that combination of IP6 and myoIns treatments reduced HOMA-IR when compared with diabetic untreated control rats [73]. IP6 also increased glucose uptake, *glucose transporter type 4* (*GLUT4*) mRNA, *insulin receptor substrate 1* (*IRS**-1*) and *phosphorylated insulin receptor substrate 1* (*p**-IRS**-1*) mRNA, decreased basal lipolysis, and increased adipocyte differentiation in 3T3L-1 mouse adipocyte [74]. This evidence suggests that IP6 plays a role in modulating insulin sensitivity in adipocytes and has anti-diabetic properties that can be mediated directly through adipocytes. IP6 can also mimic insulin effects to decrease mRNA expression and the rate of transcription of the *phosphoenolpyruvate carboxykinase* (*PEPCK*) gene, which produces an essential enzyme in gluconeogenesis, after the activation of 8-bromo-cAMP in hepatocyte of rats [75]. In addition to the transcription process, IP6 has an impact on regulation of cellular process regarding vesicle trafficking in both exocytosis and endocytosis of eukaryotic cells [2]. An experiment in insulinoma tumor HIT T15 cells (hamster islet cells) showed that IP6 stimulated pancreatic insulin exocytosis via protein kinase C (PKC pathway) [76]. IP6 also promotes pancreatic insulin endocytosis through dynamin I which activated PKC and inhibited phosphoinositide phosphatase synaptojanin pathways [77].

### 4.3. Inositol Phosphates on Obesity and Metabolic Parameters

It is well known that obesity is now considered as a major health risk globally. People worldwide are facing obesity and/or overweight status, especially in the United States. The prevalence of obesity in adults was over 30% in 2015, and is rising over time [78]. Obesity is associated not only with type 2 diabetes, metabolic syndrome, and non-alcoholic fatty liver [79], but also cancer [80]. The link between obesity and cancer was reported as an increase in leptin, insulin, IGF-1, and pro-inflammatory cytokines, which leads to an activation of phosphatidylinositol 3-kinase (PI3K)/protein kinase B (Akt) pathway and mTOR pathway, resulting in the stimulation of proliferation and survival of cancer cells [80]. So, the prevention and/or treatment of obesity should be a challenge for the good health of humankind.

The effects of inositol phosphates and IP6Ks have long been considered for a target of obesity and metabolic diseases [19]. Many experiments in mice revealed that *IP6K* knockout displayed reduced body weight, fat accumulation, and percentage of fat mass as well as increased percentage of lean body mass [21,81], suggesting the role of inositol pyrophosphates on obesity. In mice fed with a high fat diet, *IP6K* knockout also improved metabolic parameters, including blood glucose [20,21], total cholesterol, triglycerides, liver function (aspartate aminotransferase), lactate dehydrogenase (an enzyme used to detect injury of tissues), and leptin [21]. The deletion of *IP6K1* did not increase phosphorylated Akt in muscle and glycogen synthase kinase 3β (GSK3β), a rate-limiting enzyme promoting deposition of glycogen [82], in white adipose tissues which normally lead to increased impaired glucose, insulin resistance, and adipogenesis in high fat diet obese mice [21]. So, the improved insulin resistance through *IP6K1* ablation might be associated with reduced Akt pathway.

It has been known that the browning of white adipocytes leads to increased cellular thermogenesis resulting in an increase in energy expenditure [83]; this process might be one of the ways to decrease adiposity and/or obesity. Basically, the morphology of brown adipocytes are shown to be polygonal shape with multilocular lipid droplets, round nuclei, and abundant mitochondrial density, and they are considered as a key site of heat production or thermogenesis under various stimuli [84,85,86]. A study revealed that specific *IP6K1* knockout in adipocytes increased oxygen consumption rate and adipocyte browning genes [87] including *uncoupling protein 1* (*UCP1*), *peroxisome proliferator**-activated receptor gamma coactivator 1**-alpha* (*PGC1α*), *PR domain containing 16* (*PRDM16*), and *peroxisome proliferator**-activated receptor alpha* (*PPARα*), indicating that *IP6K1* deletion might enhance the browning of adipocytes [22]. Indeed, *IP6K1* deletion was also shown to increase energy expenditure and lipid oxidation via the AMP-activated protein kinase (AMPK) in mice fat cells and decreased fatty acid synthesis in 3T3-L1 adipocytes [88]. In *IP6K1* knockout mice, energy intake was not reduced, whereas body weight decreased, while oxygen consumption and energy expenditure increased, when compared with wild-type control, suggesting that IP6K1 is a dominant player in energy output [21]. In addition, adipocyte-specific *IP6K1* knockout (HFD-AdKO) mice fed with high fat diet showed improved fatty liver than their counterpart [22]. Taken together, the ablation of *IP6K1* might have the advantages in decreasing obesity, improving metabolic parameters, and increasing thermogenic energy metabolism. In other words, type 2 diabetes, obesity, and non-alcoholic fatty liver might be ameliorated by *IP6K1* deletion.

### 4.4. Inositols and Inositol Phosphates in Energy Metabolism

The association between inositols and metabolism has been linked by many research groups. Among inositol isomers, the myoIns and d-chiro-inositol have been shown to reduce risks of metabolic diseases, including diabetic mellitus, dyslipidemia [10,63,64], and PCOS [89]. d-*chiro*-inositol could be transformed from myoIns by the inversion of C3 hydroxyl via insulin-dependent epimerization [90]. Natural sources containing *chiro*-inositol include soybeans, legumes, oranges, arrowroot, and ginseng [90]. Moreover, the actions of d-*chiro*-inositol and IP6K also brought attention to the role of inositol phosphates in metabolism.

As mentioned above, 5PP-IP5 or IP7 acts as an energetic molecule and IP6K1 is a key enzyme to produce inositol pyrophosphate IP7 from IP6 [14,15]. A previous report demonstrated that IP7 levels increased during the adipogenesis of 3T3-L1 cells, and its levels were substantially reduced by IP6K1 inhibitor, suggestive of its role during the anabolic process [21]. A recent study in *db**/db* mice (type 2 diabetes mice) revealed that d-*chiro*-inositol improved glucose levels and significantly increased hepatic glycogen when compared with the control group [91]. Additionally, d-*chiro*-inositol also increased the protein expressions of insulin receptor substrate 2 (IRS2), PI3K, Akt, GLUT4, and phospho-Akt but decreased GSK3β protein in hepatic cells [91]. This finding might give a mechanism of how d-*chiro*-inositol improved glucose metabolism and hepatic glycogen synthesis through the upregulation of insulin receptor, GLUT4, GSK3β, and PI3K-Akt cascades. A previous study showed that, for mice in fasting state, glucagon stimulated PKA leading to phosphorylation of inositol 1, 4, 5-trisphosphate receptors (InsP3Rs) with an increase in Ca^2+^ in the cytosol and activation of CREB-regulated transcription coactivator 2 (CRTC2) which is associated with hepatic glucose production [92]. Whereas, for mice in fed state, hepatic glucose production might be decreased by the inactivation of InsP3Rs via the Akt pathway of the insulin action decreased CRTC2 activity [92]. These data might illustrate an important role of InsP3Rs in gluconeogenic program of the liver. Moreover, a study showed that *IP6K1* deletion might turn off the program of insulin action which activates the Akt pathway in stimulating glucose uptake, glycogen synthesis, and protein synthesis [93].

Furthermore, a previous study showed that inositol polyphosphate multikinase (IPMK) and IP6K1 played a critical role in the cascade of inositol phosphates synthesis and have an impact on energy metabolism in mammals [94]. To illustrate a simplified pathway of inositol polyphosphates synthesis, we summarize the cascade in Figure 4 and discuss the relation of pyrophosphates to energy metabolism.

For the role of inositol phosphates on lipid metabolism, deletion of *IP6K1* appeared to increase basal lipolysis by modulating protein perilipin1 (PLIN1) in 3T3L1 adipocytes suggestive that IP6K1 and inositol pyrophosphate biosynthetic process have an impact on the regulation of lipid metabolism [95]. PLIN1 is a protein coating on surfaces of adipocyte lipid droplets and serves an important function in stabilizing the lipid droplets [96]. Under the basal and hormonally stimulated lipolysis, dynamics of PLIN1 is a key factor in stimulating of lipid mobilization in adipocytes [96]. In addition to IP6K1, IP6K3 was also associated with obesity and insulin resistance regulation [97]. A previous study found that *IP6K3* mRNA in mice and humans has the highest expression in skeletal muscle when compared with other inositol kinase family members, e.g., ITPK1, IMPK, IP6K1, and IP6K2, and its expression in muscle was high in diabetic and fasting conditions of mice [97], suggestive of the association of IP6K3 with muscle glucose metabolism. *IP6K3* knockout mice showed decreased body weight, fat mass, blood glucose, blood insulin, plasma lactate, and increased glucose tolerance from age induced obesity [97]. So, IP6K3 might be another target for metabolic management. Taken together, evidence suggests that inositol phosphates have pharmacological effects on increased insulin sensitivity, improved insulin resistance and other metabolic profiles, as well as decreased obesity and adiposity. IP6K1 and inositol pyrophosphate synthesis process increased glucose, carbohydrate, and lipid metabolism. However, the molecular mechanism still needs to be clarified. Most of studies revealed that the inositol phosphates cascade plays a role in the insulin signaling pathways and also has a crucial role in the energy metabolism pathways. In addition to their effects in improving insulin resistance and lipid profiles, inositol phosphates also promote thermogenic effects of adipocytes through increased browning process of adipocytes. Most of studies focused on the effects of *IP6K1* gene ablation on alterations of metabolism in metabolic target tissues including liver and adipose tissues. However, its effects on skeletal muscle, e.g., glucose uptake and/or energy expenditure as well as molecular signaling pathways, need to be elucidated. Here, we summarize the effects of myoIns, d-*chiro*-inositol, and IP6 on metabolic alterations in Figure 5A and the effects of IP6K1 disruption on metabolic target tissues in Figure 5B, respectively. Moreover, the effects of the inositol phosphates supplement in energy metabolism and specific intracellular signaling molecules should be explored for increasing advanced knowledge and clinical implication in humans.

## 5. Other Health-Beneficial Effects

Deregulation of the inositol phosphates metabolism has been recognized in several illnesses—mostly in animal models, including neurological disorders [7], PCOS [71], metabolic diseases [10,69,70] and cancer [26,27]. Although myoIns and IP6 are prevalent natural forms and have been much studied over the last 30 years, some “other” cyclitols and inositols might also be therapeutically relevant, and their roles and applications have recently been considered [7,8,98]. For example, a study in mice with Alzheimer’s disease model examined by Morris water maze test found that 30 mg/kg of *scyllo*-inositol administered orally for 1 month improved spatial memory, and decreased plaque and amyloid-β peptides (Aβ) aggregation [99], suggestive of its therapeutic effect for cognitive deficits in Alzheimer’s disease [100]. Moreover, d-*chiro*-inositol was able to enhance the ability of insulin to protect central nervous system (CNS) synapse damage caused by the accumulation of toxic Aβ oligomers in association with Alzheimer’s disease [101]. IP6 has also been recognized as potential treatment for Alzheimer’s pathology, as evidenced from animal and in vitro models [102]. In Alzheimer’s disease, the enzyme β-secretase 1 (BACE1) and c-secretase play a key role to cleave Aβ peptides from amyloid-β precursor protein [103]. IP6 significantly inhibited BACE1 activity and reduced Aβ production in cultured SH-SY5Y neuroblastoma cells without cytotoxicity, whereas IP3, IP4, and IP5 seemed to have no effect on BACE1 activity [104].

MyoIns has been used for years against depression and anxiety disorders in patients with depressive disorder [7,105,106]. MyoIns levels were shown to be negatively correlated with levels of depression symptoms evaluated by using the Maryland Trait and State Depression (MTSD) scales [107]. Furthermore, abnormal myoIns metabolism has been shown to underlie the pathophysiology of a variety of clinical conditions including Down’s syndrome, traumatic brain injury, bronchopulmonary dysplasia (BPD), and respiratory distress syndrome (RDS) [108].

Interestingly, myoIns either alone or in combination with selenium can have beneficial effects in mice exposed to cadmium [109,110,111]. This heavy metal can be found in cigarette smoke and phosphate fertilizers, thereby possibly contaminating the environment and food sources. Exposure to cadmium can cause multiple organ damages over time due to oxidative stress. Mice exposed to cadmium were simultaneously given myoIns either alone on combination with selenium via oral routes. After 14 days, the myoIns combined with selenium group showed better markers and structural integrity than untreated mice in various organs, such as the kidneys [109], testis [110], and thyroid [111], suggesting a protective effect from oxidative stress and, thereby, a role for these compounds as nutraceuticals.

Although advanced health-beneficial effects of inositol phosphate have now been investigated, cancer preventive and therapeutic properties of IP6 have received most attention and its broad-spectrum of anticancer activities has been shown in multiple preclinical experimental studies and in humans, alone or in combination with myoIns [4,26,27,112], possibly through its involvement in the PKC pathway [112,113]. Briefly, IP6 binds to PLC coupled receptor and tyrosine kinase receptors leading to the activation of phosphoinositide-specific phospholipase C (PI-PLC) to cleave PIP2 into InsP3 and hydrophobic sn-1,2-diacylglycerol (DAG) which will activate PKC.

Another interesting line of studies indicated that the consumption of IP6 can prevent development of osteoporosis and had a protective effect against osteoporosis [114]. IP6 was also shown to inhibit osteoclast bone resorption induced by receptor activator of nuclear factor kappa-Β ligand (RANKL) in primary osteoclasts of humans [115]. It has been known that adipocytes and osteocytes in the bone marrow are differentiated from bone marrow-derived mesenchymal stem/stromal cells (BMMSCs) [116]. An experiment in MSCs isolated from young mice (2 months old) showed that *IP6K1* knockout BMMSCs reduced adipocyte differentiation and enhanced osteocytes [116]. This evidence might suggest the important role of inositol phosphates on osteoporosis prevention or age related bone diseases.

Furthermore, anti-inflammatory effects of IP6 and myoIns was shown [117]; IP6 reduced hepatocellular necrosis and pro-inflammatory cytokines mRNA including *tumor necrosis factor alpha*
*(TNF**-α**)*, *interleukin 6*
*(IL**-6**)*, *and interleukin**-I beta*
*(IL**-1β**)* in liver tissues of iron overloaded induced liver injury mice [118]. d-*myo*-inositol supplement both orally (400 mg/kg) and intraperitoneally (200 and 800 mg/kg) suppressed inflammation in rats with adjuvant disease [119]. Moreover, *myo*-inositol-1-phosphate synthase (Ino-1) deletion in bacteria (*Actinobacteria Corynebacterium glutamicum*) increased production of ROS and decreased cell viability suggested that Ino-1 might have an effect on oxidative stress resistance [120]. Because of its ability to downregulate inflammation and cytokine release, it has been recently speculated that myoIns might be beneficial for Sars-CoV-2 patients [121].

Moreover, the prevention of kidney stones and other pathological calcifications, such as sialolithiasis, a common disease of salivary glands, and cardiovascular calcification, that frequently occurs in the heart vessels, has been known for IP6 [24]. Moreover, based on various clinical evidence, the use of myoIns in humans was revealed to be safe [122], but some side effects had been observed such as flatus, diarrhea, and nausea; however, the adverse effects did not increase with the increased dosage [123]. Very recently, the clinical use of myoIns and d-*chiro*-inositol, both insulin-sensitizing agents, in assisted reproductive treatment (ART) has been suggested, because of the known beneficial effects of these inositols in both female and male reproduction [124].

## 6. Conclusions and Future Research

Indeed, inositols and the inositol phosphates have been shown to have diverse health benefits such as anticancer, anti-diabetic, anti-oxidant, and anti-inflammation [4,27,69]. Much of the evidence discussed herewith have shown its therapeutic effects on various diseases and conditions, suggesting their diverse properties can promote human health and their possible role as nutraceuticals. Interestingly, in the last several years, much attention has been given to the role of inositols and inositol phosphates in metabolic disorders. As mediators of insulin action, myoIns, d-*chiro*-inositol, and IP6 have been shown to consistently decrease insulin resistance and to improve insulin sensitivity. However, the effects of inositol phosphates on carbohydrate and glucose metabolism are still poorly understood.

The complex interaction among inositol molecules, insulin signaling, and carbohydrate and/or glucose metabolism pathways would be intriguing for further studies. We would hypothesize that inositol phosphates might alter the actions of second messenger molecules in the energy metabolism pathways of insulin-sensitive tissues including liver, muscle, and adipose tissue. The better understanding on the interplay of these molecules in insulin-sensitive tissues might explore novel knowledge and provide treatment options to treat patients with metabolic syndrome. Interestingly, discussing biological regulation systems for metabolic networks classification, when a new metric system is proposed for comparisons of different metabolic systems, inositol and its derivatives involved in membrane signaling are featured as a simple molecular system offering new perspectives [125].

Lastly, because inositols and the inositol phosphates, are abundant in food sources, especially in the Mediterranean diet [69], their roles as nutraceuticals become increasing attractive. It would be intriguing to observe some of their other health benefits in clinical settings. Specifically, an initial study has been reported on a cohort of 185 mother–infant pairs to assess whether a Mediterranean diet during pregnancy may modify the birth outcome of prenatal cadmium exposure at non-occupational levels [126]. Although larger trial may be necessary to elucidate their roles, it is a positive step towards our current understanding in the field.

## Figures and Tables

**Figure 1 molecules-25-05079-f001:**
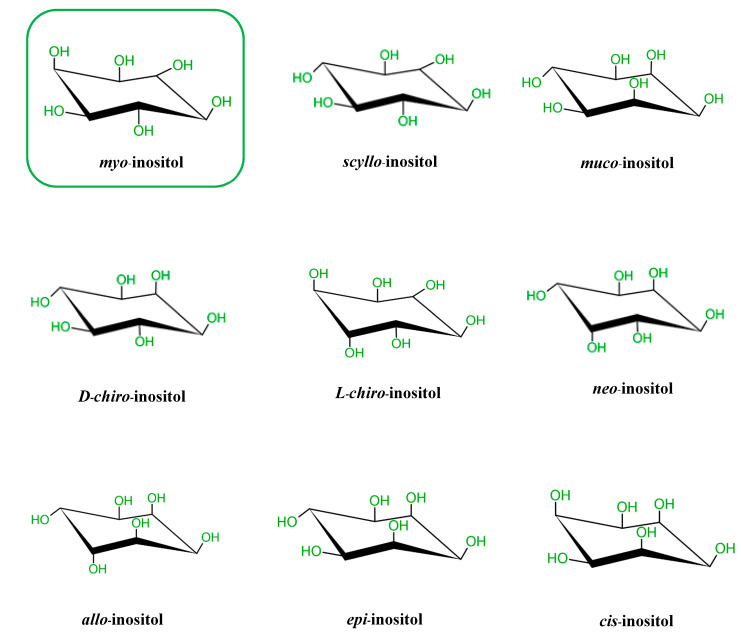
Structures of the 9 stereoisomers of inositol, which exist under 9 stereoisomeric forms through epimerization of its hydroxyl groups. *Myo*-Inositol (framed) is the most common isomer in plants and animal cells.

**Figure 2 molecules-25-05079-f002:**
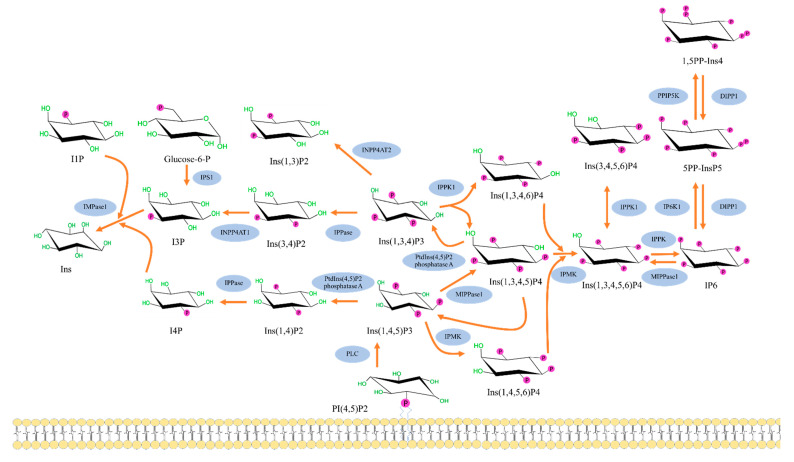
The inositol polyphosphates pathway in human cells. The figure shows metabolism cascade of inositol phosphate molecules from the well-known Ins (1,4,5) P3 formation via phospholipase C to higher inositol phosphates, including pyrophosphates (5PP-IP5 or IP7 and I,5PP-IP4 or IP8). In contrast, inositol triphosphates Ins (1,3,4) P3 and Ins (1,4,5) P3 can be broken down into lower inositol phosphates and eventually myoIns. DIPP1: Diphosphoinositol polyphosphate phosphohydrolase 1; IMPase 1: Inositol monophosphatase 1; INPP4AT1: Type I inositol 3,4-bisphosphate 4-phosphatase; INPP4AT2: Type II inositol 3,4-bisphosphate 4-phosphatase; IP3K A: Inositol-trisphosphate 3-kinase A; IP6K1: Inositol hexakisphosphate kinase 1; IPMK: Inositol polyphosphate multikinase; IPPase: Inositol polyphosphate 1-phosphatase; IPPK: Inositol-pentakisphosphate 2-kinase; IPS1: Inositol-3-phosphate synthase 1; ITPK1: Inositol-tetrakisphosphate 1-kinase; MIPPase1: Multiple inositol polyphosphate phosphatase; PPIP5K: Inositol hexakisphosphate and diphosphoinositol-pentakisphosphate kinase 5; PtdIns (4,5) P2: Phosphatidylinositol 4,5-bisphosphate 5-phosphatase A [18].

**Figure 3 molecules-25-05079-f003:**
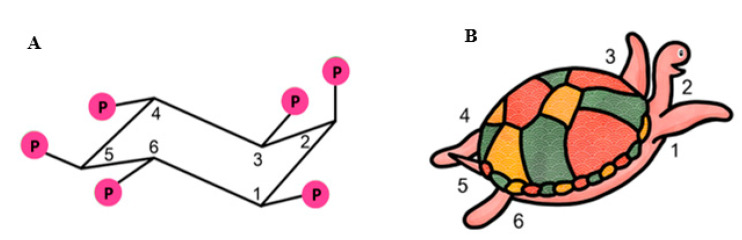
The structure of IP6 and the turtle analogy. Panel (**A**) shows chair conformation of *myo*-inositol hexakisphosphate (IP6) with the unique configuration of phosphate groups in positions 1, 2 and 3 (axial-equatorial-axial). Panel (**B**) shows Agranoff’s turtle analogy.

**Figure 4 molecules-25-05079-f004:**
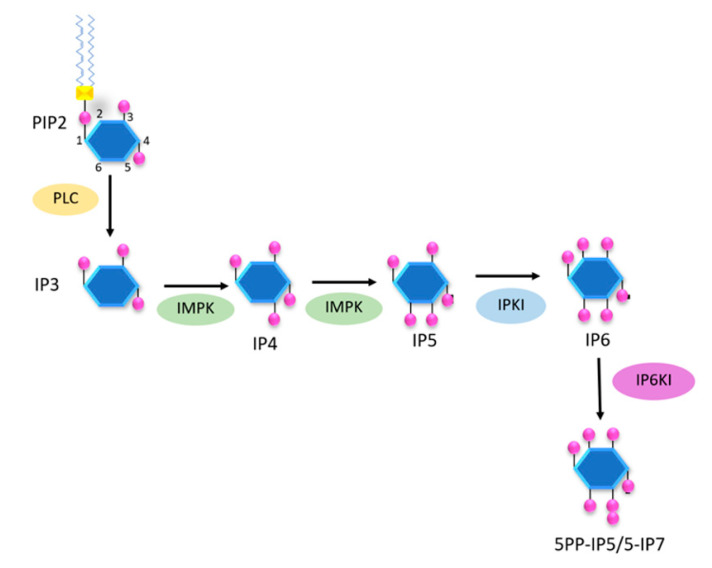
The basic pathway of inositol polyphosphates involved in energy metabolism. The figure shows an oversimplified pathway from PIP2 to pyrophosphates, with IP6 having a central role and position.

**Figure 5 molecules-25-05079-f005:**
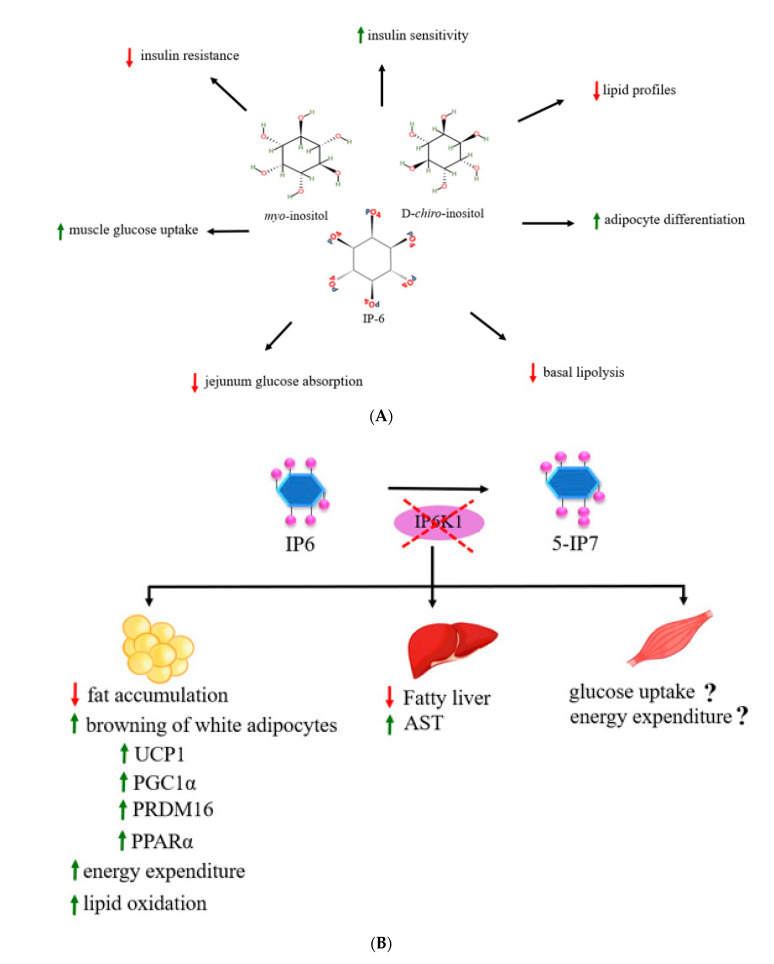
The diagram shows the effects of myoIns, d-*chiro*-inositol, and IP6 on insulin metabolism, glucose metabolism, and other metabolic profiles (**A**) and the effects of IP6K1 disruption in metabolic target tissues including adipose tissues, liver, and skeletal muscle (**B**).

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
