# Peer review of "Role of Inositols and Inositol Phosphates in Energy Metabolism"

_molecules, 2020, doi:10.3390/molecules25215079_

Round 1
Reviewer 1 Report
In the present review paper, S. Chatree and coworkers examine the inositol phosphate metabolism in mammalian cells with the aim to establish their biological activities and highlight their potential roles in energy metabolism. The authors indicated that, despite all recent advances, still many aspects of the bioactivity of inositol phosphates are not clear, especially their effects on insulin resistance and alteration of metabolism. Overall, I think that the paper is nice and timely, and it will be of great interest to the readers and researchers, in general. I make some suggestions for further improve the quality of the review.
There is recent evidence that myo-inositol, in combination with Selenium, showed a benefit against thyroid, kidney and testis alterations caused by exposure to Cadmium, also unmasking a new mechanism of action of inositol. So, considering these experimental observations, myo-inositol might offer a new possible nutraceutical challenge that, properly combined with good agricultural practice to minimize Cd contamination in food crops and animals, could also provide a definite strategy in the fascinating field of research investigating interactions between Molecules and environment. Please discuss these experimental results (also from a translational perspective) and kindly include related references (i.e. Benvenga S, et al. Curr Mol Pharmacol. 2019;12(4):311-323.; Pallio G. et al. Food Chem Toxicol. 2019 Oct; 132:110675; Benvenga S. et al. Nutrients. 2020 Apr 26;12(5):1222) in your review paper, focusing this key point particularly in the sub-section 5. “Other Health-Beneficial Effects” and in the Conclusions.
Author Response
Response to the reviewers
Reviewer(s)' Comments to Author:
Reviewer 1
Reviewer’s Comment 1: There is recent evidence that myo-inositol, in combination with Selenium, showed a benefit against thyroid, kidney and testis alterations caused by exposure to Cadmium, also unmasking a new mechanism of action of inositol. So, considering these experimental observations, myo-inositol might offer a new possible nutraceutical challenge that, properly combined with good agricultural practice to minimize Cd contamination in food crops and animals, could also provide a definite strategy in the fascinating field of research investigating results (also from a translational perspective) and kindly include related reference (i.e. Benvenga S, et al. Curr Mol Pharmacol 2019; 12(4):311-323., Pallio G, et al. Food Chem Toxicol. 2019 Oct; 132:110675; Benvenga S. et al. Nutrients. 2020 Apr 26;12(5): 1222) in your review paper, focusing this key point particularly in the sub-section 5. “Other Health-Beneficial Effects” and in the Conclusions.
Authors’ Response 1:
- We would like to offer our gratefulness and deep appreciation for the reviewer’s review and suggesting additional references. We have included the segment on inositol and selenium in lines 480-488, page 26 as follows:
Interestingly, myoIns either alone or in combination with selenium can have beneficial effects in mice exposed to cadmium [109-111]. This heavy metal can be found in cigarette smoke and phosphate fertilizers, thereby possibly contaminating the environment and food source. Exposure to cadmium can cause multiple organ damages over time due to oxidative stress. Mice exposed to cadmium were simultaneously given myoIns either alone on combination with selenium via oral routes. After 14 days, the myoIns combined with selenium group showed better markers and structural integrity than untreated mice in various organs, such as the kidneys [109], testis [110], and thyroid [111], suggesting a protective effect from oxidative stress and, thereby, a role for these compounds as nutraceuticals.
- We also revised the conclusions and future research section to include recent clinical study on Mediterranean diet and maternal cadmium exposure in line 544-550, page 29 as follows:
Lastly, because inositols and the inositol phosphates, are abundant in food sources especially Mediterranean diet [69], their roles as nutraceuticals become increasing attractive. It would be intriguing to observe some of their other health benefits in clinical settings. Specifically, an initial study has been reported on a cohort of 185 mother-infant pairs to assess whether Mediterranean diet during pregnancy may modify the birth outcome of prenatal cadmium exposure at non-occupational levels [126]. Although larger trial may be necessary to elucidate their roles, it is a positive step towards our current understanding in the field.
- Additional References that were included:
- 69. Dinicola, S.; Minini, M.; Unfer, V.; Verna, R.; Cucina, A.; Bizzarri, M., Nutritional and Acquired Deficiencies in Inositol Bioavailability. Correlations with Metabolic Disorders. Int J Mol Sci 2017, 18, (10).
- 109. Pallio, G.; Micali, A.; Benvenga, S.; Antonelli, A.; Marini, H. R.; Puzzolo, D.; Macaione, V.; Trichilo, V.; Santoro, G.; Irrera, N.; Squadrito, F.; Altavilla, D.; Minutoli, L., Myo-inositol in the protection from cadmium-induced toxicity in mice kidney: An emerging nutraceutical challenge. Food Chem Toxicol 2019, 132, 110675.
- 110. Benvenga, S.; Micali, A.; Pallio, G.; Vita, R.; Malta, C.; Puzzolo, D.; Irrera, N.; Squadrito, F.; Altavilla, D.; Minutoli, L., Effects of Myo-inositol Alone and in Combination with Seleno-Lmethionine on Cadmium-Induced Testicular Damage in Mice. Curr Mol Pharmacol 2019, 12, (4), 311-323.
- 111. Benvenga, S.; Marini, H. R.; Micali, A.; Freni, J.; Pallio, G.; Irrera, N.; Squadrito, F.; Altavilla, D.; Antonelli, A.; Ferrari, S. M.; Fallahi, P.; Puzzolo, D.; Minutoli, L., Protective Effects of Myo-Inositol and Selenium on Cadmium-Induced Thyroid Toxicity in Mice. Nutrients 2020, 12, (5).
- 126. Gonzalez-Nahm, S.; Nihlani, K.; J, S. H.; R, L. M.; H, G. S.; Hoyo, C., Associations between Maternal Cadmium Exposure with Risk of Preterm Birth and Low after Birth Weight Effect of Mediterranean Diet Adherence on Affected Prenatal Outcomes. Toxics 2020, 8, (4).

Reviewer 2 Report
The article “Role of Inositol Phosphates in Energy Metabolism” summarizes the inositol phosphate metabolism in mammalian cells, discussing their biological activities and the potential roles in energy metabolism. These molecules – myo-Ins and phosphate derivatives - are known to decrease insulin resistance, increase insulin sensitivity, and display several modulatory effects upon metabolism. The subject is extensively reviewed and properly discussed, by referring exhaustively to a relevant body of scientific literature.
Minor observation
- The text should be checked for some imprecisions and misspellings. For instance, in the abstract “inositol phosphates, especially myo-inositol and inositol hexaphosphate, also known as phytic acid or IP6”, should be emended: myo-inositol is not an “inositol phosphate” and IP6, accordingly to IUPAC nomenclature, is properly recognized as “inositol hexakisphosphate” and not “hexaphosphate”.
- Again, line 265, “D-chiro-inositol” is not a phosphate derivative of inositol, instead it is an epimer of myo-Ins
- In some cases, I do not understand the use o italics, es. Line 297 “phosphoenolpyruvate carboxykinase (PEPCK)”
Author Response
Response to the reviewers
Reviewer(s)' Comments to Author:
Reviewer 2
Reviewer’s Comment 1:
Minor observation
The text should be checked for some imprecisions and misspellings. For instance, in the abstract “inositol phosphates, especially myo-inositol and inositol hexaphosphate, also known as phytic acid or IP6”, should be emended: myo-inositol is not an “inositol phosphate” and IP6 accordingly to IUPAC nomenclature, is properly recognized as “inositol hexakisphosphate” and not hexaphosphate.
Authors’ Response 1:
- We thank the reviewer for your meticulousness. We have now revised the abstract as the reviewer comment in line 24, page 2 as follows:
Recently, inositol, especially myo-inositol and inositol hexakisphosphate, also known as phytic acid or IP6, with their biological activities received much attention for their role in multiple health beneficial effects.
- Furthermore, we have now corrected “inositol hexaphosphate” to “inositol hexakisphosphate” in the text.
- We also changed the title of the manuscript for a better representation of the main idea in line 1, page 1 as follows:
Role of Inositols and Inositol Phosphates in Energy Metabolism
- We also corrected the subheading for the consistency in line 56, page 3; in line 267, page 13; and line 361, page 17 as follows:
- 2. Biochemistry of Inositols and Inositol Phosphates
4.2 Inositol, Inositol Phosphates and Insulin Resistance
4.3 Inositols and Inositol Phosphates on Energy Metabolism
Reviewer’s Comment 2:
Again, line 265, “D-chiro-inositol” is not a phosphate derivative of inositol, instead it is an epimer of myo-Ins.
.
Authors’ Response 2: We thank the reviewer for the kind comment and recommendation. We have now revised the sentence in lines 271-272, page 15 as follows:
For example, studies found that myoIns and D-chiro-inositol have been shown to counteract metabolic disorders [63] and improved insulin resistance [63-65].
Reviewer’s Comment 3:
In some cases, I do not understand the use italics, es. Line 297 “phosphoenolpyruvate carboxykinase (PEPCK)”
Authors’ Response 3:
- We thank the reviewer for the comment. We use italics to represent the gene expression of the whole manuscript. We have now added the word “gene” for a better explanation in line 310-311, page 15 as follows:
IP6 can also mimic insulin effects to decrease mRNA expression and the rate of transcription of phosphoenolpyruvate carboxykinase (PEPCK) gene, which produces an essential enzyme in gluconeogenesis, after activation of 8-bromo-cAMP in hepatocyte of rats [75].
- We also corrected all the abbreviations annotating genes to italics in lines 323-354, page 17-19. For example, in line 332-334:
“So, the improved insulin resistance through IP6K1 ablation might be associated with reduced Akt pathway.”
IP6K1 is now italicized to indicate a gene, whereas Akt remains normal to indicate protein.
References
- 63. Minozzi, M.; Nordio, M.; Pajalich, R., The Combined therapy myo-inositol plus D-Chiro-inositol, in a physiological ratio, reduces the cardiovascular risk by improving the lipid profile in PCOS patients. Eur Rev Med Pharmacol Sci 2013, 17, (4), 537-40.
- 64. Mancini, M.; Andreassi, A.; Salvioni, M.; Pelliccione, F.; Mantellassi, G.; Banderali, G., Myoinositol and D-Chiro Inositol in Improving Insulin Resistance in Obese Male Children: Preliminary Data. Int J Endocrinol 2016, 2016, 8720342.
- 65. Bevilacqua, A.; Bizzarri, M., Inositols in Insulin Signaling and Glucose Metabolism. Int J Endocrinol 2018, 2018, 1968450.
- 75. Alvarez, L.; Avila, M. A.; Mato, J. M.; Castaño, J. G.; Varela-Nieto, I., Insulin-like effects of inositol phosphate-glycan on messenger RNA expression in rat hepatocytes. Mol Endocrinol 1991, 5, (8), 1062-8.
